# Agricultural Weed Assessment Calculator: An Australian Evaluation

**DOI:** 10.3390/plants9121737

**Published:** 2020-12-09

**Authors:** Hugh J. Beckie, Mechelle J. Owen, Catherine P.D. Borger, Gurjeet S. Gill, Michael J. Widderick

**Affiliations:** 1Australian Herbicide Resistance Initiative, School of Agriculture and Environment, The University of Western Australia, Perth 6009, Australia; mechelle.owen@uwa.edu.au; 2Department of Primary Industries and Regional Development, Northam 6401, Australia; catherine.borger@dpird.wa.gov.au; 3Discipline of Agricultural and Animal Science, The University of Adelaide, Adelaide 5064, Australia; gurgeet.gill@adelaide.edu.au; 4Queensland Department of Agriculture and Fisheries, Toowoomba 4350, Australia; michael.widderick@daf.qld.gov.au

**Keywords:** agricultural weed, herbicide resistance, weed abundance, weed competition, weed loss, weed risk assessment

## Abstract

Weed risk assessment systems are used to estimate the potential weediness or invasiveness of introduced species in non-agricultural habitats. However, an equivalent system has not been developed for weed species that occur in agronomic cropland. Therefore, the Agricultural Weed Assessment Calculator (AWAC) was developed to quantify the present and potential future adverse impact of a weed species on crop production and profitability (threat analysis), thereby informing or directing research, development, and extension (RDE) investments or activities. AWAC comprises 10 questions related primarily to a weed’s abundance and economic impact. Twenty weed species from across Australia were evaluated by AWAC using existing information and expert opinion, and rated as high, medium, or low for RDE prioritization based on total scores of 70 to 100, 40 to <70, or <40, respectively. Five species were rated as high (e.g., *Lolium rigidum* Gaud.), eight were rated as medium (e.g., *Conyza* spp.), and seven were rated as low (e.g., *Rapistrum rugosum* L.). Scores were consistent with the current state of knowledge of the species’ impact on grain crop production in Australia. AWAC estimated the economic or agronomic threat of 20 major or minor agricultural weeds from across Australia. The next phase of development is the testing of AWAC by weed practitioners (e.g., agronomists, consultants, farmers) to verify its utility and robustness in accurately assessing these and additional weed species.

## 1. Introduction

The Australian weed risk assessment system [1] and systems adapted from it have been widely used to evaluate the possible risk of adverse impacts from introductions of alien plants [2]. The system comprises 49 questions related to the species’ biogeography, undesirable attributes, biology, and ecology. Species with an overall score of less than one are rated as acceptable for introduction, those scoring greater than six are rejected, and those scoring between one and six require further evaluation. This system has been highly successful in Australia, with recent economic analysis suggesting use of the system was profitable within a decade and will save up to AUD 1.8 billion over 50 years, due to exclusion of species posing a biosecurity risk [3]. The system has further proven effective across a range of geographical regions outside Australia and New Zealand, the countries for which it was developed, including Hawaii, Czech Republic, Bonin Islands, and Florida [4]. The United States Department of Agriculture adopted a different assessment system, which improved the accuracy in identifying non-invaders [5,6].

To complement the Australian weed risk assessment system, the national post-border weed risk management protocol comprises 24 questions, of which six are relevant to agriculture: (1) impact yield?; (2) impact quality?; (3) affect land value?; (4) change land use?; (5) increase harvest costs?; (6) disease host/vector? (qualitatively rated as low, low to moderate, moderate to high, high) [7]. For example, the states of Victoria and New South Wales use this weed risk assessment impact criteria for recently introduced species.

Regardless of the decision-support system used by governments, their focus is on determining the potential weediness or invasiveness (i.e., biosecurity risk) of plant species newly introduced to an ecological system. They are not designed to determine present or future agronomic and economic risk posed by existing species. An existing weed species may be widely distributed throughout an agricultural region, placing it beyond biosecurity issues of containment or eradication (with notable exceptions such as *Chondrilla juncea* L. in Western Australia [8]). For these species, existing weed risk assessment tools are not applicable. Even though a species may be common, this status does not guarantee that we fully understand the risk it poses to industry. An example is *Bromus* spp. (brome grass), which have been common weeds in Australia since the 1800s but have increased in economic impact to become our fourth most problematic weed in recent decades [9]. A weed risk assessment tool tailored to existing agricultural weeds rather than potential new threats to the system, and which quantifies their present and potential future adverse impact on crop production and profitability (threat analysis), can help inform, prioritise, and direct public-sector (e.g., local, state/provincial/national government departments and agencies; crop commodity research and development boards/commissions; non-profit organizations) or private-sector (e.g., agrochemical companies; consultants) research, development, and extension (RDE) investments, or activities to address and mitigate the predicted threat [10].

Accordingly, the Agricultural Weed Assessment Calculator (AWAC) described herein was developed and subsequently evaluated with 20 major or minor weed species across the western, southern, and northern grain-cropping regions of Australia [11]. The western region is synonymous with the state of Western Australia, the southern region with the states of South Australia, Victoria and Tasmania, and the northern region with New South Wales and Queensland. The geographic centre of origin and current distribution and abundance of these species in Australia are briefly described in the remainder of this section.

In the last 40 years, *L. rigidum* Gaud. (annual or rigid ryegrass) has been the most important weed in agronomic field crops. It ranks first among weed species in relative abundance (based on field frequency and plant density), although less prevalent in the northern than southern or western regions (national and regional weed species abundance ranking compiled in Llewellyn et al. [9]). *Lolium rigidum* is native to the Mediterranean region and was introduced into Australia in the late 1800s as a pasture species [12]. With a shift to continuous cropping in the 1980s, it became a major, economically damaging crop weed. *Raphanus raphanistrum* L. (wild radish), native to Europe, is an important annual dicot weed infesting crops in many parts of the world [13]. Averaged across the grain-growing regions of Australia, *R. raphanistrum* ranks second in abundance, being more prevalent in the western and southern regions than the northern region similar to that of *L. rigidum*. *Avena* spp. (*A. fatua* L. and *A. sterilis* L.; collectively wild oats) co-occur in Australia. However, *A. sterilis* (sterile oat or winter wild oat) tends to be more prevalent in warmer areas of northern New South Wales and southern Queensland, while *A. fatua* dominates in the western and south-eastern areas [14]. *Avena* spp. rank third in abundance, being more prevalent in the northern and southern regions than the western region. The centre of origin of *Avena* spp. is the western Mediterranean region [15].

*Hordeum* spp. (*H. leporinum* Link and *H. glaucum* Steud.) (collectively barley grass) is the ninth most abundant weed across Australia, occurring predominantly in the western and southern regions. Despite its high relative abundance, it is categorized as an “emerging” weed species, i.e., increasing abundance or range over time [16]. These *Hordeum* spp. are native to the Mediterranean region or south-eastern Europe [17]. *Bromus* spp. (*B. diandrus* Roth and *B. rigidus* Roth) (collectively brome grass) is the fourth most abundant weed across Australia, being most common in the western and southern regions. Similar to *Hordeum* spp., it is recognized as an important emerging weed species with increasing abundance and range expansion over the past 10 years [18]. This species also originates from the Mediterranean region [19,20].

*Arctotheca calendula* (L.) Levyns (capeweed) ranks 10th nationally in abundance and is more prevalent in the southern and western regions than the northern region. It was introduced into Western Australia from South Africa [19]. Although traditionally considered a pasture or environmental weed, it has become a common weed of cropland. *Emex* spp. (*E. spinosa* (L.) Campd. and *E. australis* Steinh. syn. *Rumex hypogaeus*; doublegee) is the 11th most abundant crop weed nationally, ranging from 9th place in the western region to 16th place in the northern region. Similar to *A. calendula*, it was introduced into Australia from South Africa [21]. *Emex* spp. was traditionally considered a minor weed but is now recognized as an emerging weed [16].

*Chloris virgata* Sw. (Feathertop Rhodes grass) is a tufted annual species originating from the tropical Americas. It is well established in the northern grain region of Australia (ranked 8th in relative abundance, 18th nationally) and is emerging as a problem in the southern region [22]. Considered an invasive and environmental weed, it is now becoming an important agricultural weed.

*Echinochloa* spp. (*E. colona* (L.) Link and *E. crus-galli* (L.) P. Beauv.) (collectively barnyard grass), native to Asia, rank as the 16th most abundant weed species nationally (10th most abundant weed in the northern region). *Echinochloa colona* occurs throughout the eastern and northern regions, whereas *E. crus-galli* is primarily found in the southern and eastern regions [14]. These summer species are favoured in reduced-tillage systems and have increased in prevalence in the last two decades [23].

*Sonchus oleraceus* L. (sowthistle) is the eighth most abundant weed in Australia, ranging from second place in the northern region to 17th place in the southern region. It is native to Europe, Northern Africa, and Western Asia [24]. In the northern region, the weed is common in both crop and fallow, but most prevalent in fallow [14]. It has become more common over the past 30 years in all regions, partly due to increased adoption of no-tillage cropping systems [25,26].

*Conyza* (syn. *Erigeron*) spp. (e.g., *C. bonariensis* (L.) Cronq., *C. sumatrensis* (Retz.) E. Walker, *C. canadensis* (L.) Cronq.) (collectively fleabane), native to the Americas, occur in all grain-cropping regions of Australia and rank seventh in abundance nationally. In this weed genus, *C. bonariensis* (flaxleaf fleabane) is the most abundant and widespread species in grain-cropping regions [9]. Similar to *S. oleraceus*, the weed occurs mainly in the northern region and is considered an emerging weed in the southern and western regions [27]. The biology and ecology are similar to *S. oleraceus*, particularly in two aspects: (1) shallow soil-germinating weed species well adapted to no-tillage regimes; (2) seeds are easily and widely dispersed by wind [14].

*Lactuca serriola* L. (prickly lettuce), originating from Eurasia, is most abundant in the southern region (ranked 18th in abundance). It is a common weed of fallow and non-crop disturbed (ruderal) areas but can be a troublesome weed in cereals and pastures [28]. Another abundant weed in the southern region is *Sisymbrium orientale* L. (Indian hedge mustard or wild mustard), although it occurs in all regions and ranked sixth overall (ranging from fourth in the southern region to 13th in the western region). Originating from Eurasia and North Africa, it commonly occurs in cropland, pastures, and ruderal areas [14].

*Fallopia convolvulus* L. Á. Löve (syn. *Polygonum convolvulus*) (black bindweed), a weed species native to Eurasia, is ranked 17th in relative abundance nationally; however, it is commonly found in the northern region (seventh place ranking). *Polygonum* spp. (*P. aviculare* L. and *P. arenastrum* Boreau) (collectively wireweed), native to Eurasia, ranks 13th in relative abundance and occurs in all three grain-cropping regions. It is considered an emerging weed because of increased abundance and distribution over the past decade [16]. *Moorochloa eruciformis* (Sm.) Veldkamp (sweet summer grass), native to Africa, Mediterranean region, and Asia, does not rank in the top 20 weed species nationally but is the 13th most abundant weed in the northern region. Its occurrence and importance have increased over the past 20 years with increased adoption of no-tillage cropping [14]. *Rapistrum rugosum* (L.) All. (turnip weed or wild turnip), native to Europe, is commonly found in all three regions, ranked fifth overall. *Amsinckia* spp. (yellow burr weed), native to the Americas, occurs mainly in the southern region (ranked 13th); it is categorized as a “declared pest” in the western region, with associated biosecurity measures for its containment [29]. Another declared pest in this region is *Chondrilla juncea* L. (skeleton weed), a species native to Eurasia that occurs in southern Australia. *Vicia* spp. (vetches) rank 12th on a national basis and are also commonly found in the southern region.

The objectives of this study are to describe and assess the utility of AWAC in estimating the present and potential future adverse impact of various weed species on annual grain crop production and profitability. AWAC comprises 10 questions related primarily to a weed’s abundance and economic impact. Twenty weed species from across Australia, as outlined above, were evaluated using existing information and expert opinion.

## 2. Materials and Methods

The desired criteria for AWAC were simplicity, brevity (10 questions maximum), general availability of information to address each question, and comprising the essential aspects of the weed species (e.g., biology, population abundance), crop-weed interactions, or its management in grain crop production. The calculator was developed prior to individual weed species being evaluated to avoid potential bias. The AWAC can be applied locally, regionally (e.g., western, southern, northern crop production regions), or nationally (amalgamated across agroregions). Winter cropping (April to October) dominates in the western and southern regions of Australia with land commonly fallowed in the hotter, drier summer period (November to March), whereas both winter and summer crops are commonly grown in the northern region [11].

The AWAC is described in Table 1. The first two questions are related to the weed species current relative abundance and trends over time in relative abundance, respectively, as detailed in the previous section for the 20 species to be evaluated with AWAC. A weed species often falls into one but not both of these categories. The designated region can include the western, southern, and/or northern grain-cropping areas of Australia. Weed abundance information may be derived from multiple sources, such as field and grower surveys or herbarium data [30]. Some weeds may not be abundant or economically important on a national basis but can have a significant local or regional impact due to myriad factors such as habitat preference or climatic suitability. Field surveys often quantify residual weed species abundance, i.e., plants escaping weed management practices that are usually herbicide-based. Accordingly, weed species abundance and trends over time in an agro-ecoregion are primary criteria for prioritizing public- or private-sector applied research, or assessing weeds for risk of herbicide resistance [10,31]. It is important to understand why weed species are increasing over time in order to develop effective control strategies and tactics.

Questions three and four relate to the availability of herbicides to control the weed in the major crops grown in the region or during the non-crop phase such as pasture or fallow land. Information to address these questions were derived from Anonymous [32] or a herbicide’s registration label available online. The degree of control by herbicides is a question also posed in the Australian weed risk assessment system [1]. For example, newly introduced crops may not have adequate herbicide options to control problematic weed species. Even for established crops, new weed species moving into an agricultural region may require the registration of additional herbicides for their control. If there are sufficient herbicide tools to control the weed species, then there is less threat to crop production and less incentive for RDE investment or activities. The future availability of some key herbicides is uncertain, due to increasing pesticide regulatory restrictions, requirements, or costs, social license pressures and judicial litigations, or herbicide resistance [33].

Clearly, the extent of resistance in a species to different herbicide sites of action (SOA) may compromise the efficacy of one or more herbicides (question 5). Populations of the majority of agricultural weed species in Australia exhibit varying degrees of herbicide resistance, particularly to acetyl-CoA carboxylase (ACCase) and acetolactate synthase (ALS) inhibitors (e.g., see Owen et al. [34,35]). Weed populations that are resistant to multiple herbicide SOA due to enhanced metabolism or sequential selection are being increasingly reported [36]. These biotypes threaten the effectiveness of existing and even yet-to-be commercialized herbicides (e.g., see Brunton et al. [37]). Weed population size (questions 1 and 2) directly affects the risk of herbicide resistance because of greater herbicide selection pressure and naturally occurring de novo mutations conferring resistance.

A primary consideration in any weed risk assessment tool is the potential revenue loss caused by the weed’s interference in crop productivity or quality and the herbicidal cost of control (questions 6, 8, and 10, respectively). As net returns or gross margins in crop production are calculated as grain commodity revenue (yield x price) minus variable (e.g., herbicide) costs, profitability is directly impacted by either parameter. Herbicide costs were derived from Anonymous [32]. Control of grass weeds in cereal crops typically is more expensive than that of broadleaf weeds. Some weeds, such as *R. raphanistrum*, typically require two in-season herbicide applications for effective control, which increases both application costs and herbicide product costs. Weed species vary in their competitive ability. For example, crop competitiveness of *Bromus* spp. is significantly greater than that of *S. oleraceus* [16]. Crop yield loss will likely occur if weeds are not controlled during their “critical period of weed control”, i.e., weed growth stages resulting in little crop yield loss. Fortunately, there are numerous studies published in the literature over the past 50 years on the effect of density or relative time of emergence of various weeds on seed yield loss of various crops.

For a weed to be classified as a grain contaminant, two common criteria are ease and cost of removing weed seeds from crop seedlots and mammalian toxicity. Toxicity to animals is considered as an undesirable trait in the Australian weed risk assessment system [1]. Some weeds, such as *Sinapis arvensis* L., have seeds both difficult to separate from crop seed because of similar shape and size (e.g., oilseed rape, *Brassica napus* L.) and a maximum tolerance in oilseed rape seedlots (ca. 5%) because of linolenic and erucic acid levels impacting crop oil and meal composition [38]. Moreover, high levels of glucosinolates in seeds can cause serious illness in livestock.

A score of 10 points for question 9—weed species considered troublesome, i.e., economically damaging or difficult and costly to control—should be in agreement for points allocated to any of questions 6, 8, or 10. If farmers, land managers, or weed practitioners do not consider the weed as troublesome, then the importance of any adverse economic and agronomic impact caused by the weed may be questionable. Farmer perception of the most serious weed problems do not always coincide with field research results [10]. Accordingly, grower surveys and field research are both important and complementary.

Question 7 addresses the potential for propagule (seeds, pollen, rhizomes/stolons) dispersal, analogous to questions in the Australian weed risk assessment system related to dispersal mechanisms [1]. This weediness or invasiveness characteristic not only impacts population growth rate and range expansion, but also the spread of herbicide resistance alleles across the landscape [39]. Efficient pollen and seed dispersal are best exemplified by *Kochia scoparia* (L.) Schrad., an introduced species to North America with one of the fastest rates of spread [40,41]. Although introduced into Australia in 1990 as a forage, *K. scoparia* was successfully eradicated [42].

The highest possible total score for a weed species is 100 points (Table 1). Before 20 major and minor agricultural weed species were evaluated, the level of RDE prioritization was categorized as high, medium, or low based on total scores of 70 to 100 points, 40 to <70 points, and <40 points, respectively. This qualitative categorization for each weed species gave some context to their quantitative score, although the latter is most important. The individual score for each question was based on the literature and unanimous consensus among the authors and following external feedback from some weed scientists. The 20 weed species evaluated were selected according to their relative abundance or revenue loss ranking nationally or regionally [9]. Major and minor species were selected to cover the range in magnitude of these two parameters. They were selected to provide an expected range of potential RDE priorities (i.e., total scores) following evaluation by AWAC. Therefore, the strategy for this evaluation was to allow for a robust comparison of both major and minor or emerging weed species to best assess the versatility, utility, and accuracy of the decision-support system.

## 3. Results and Discussion

For the 20 evaluated weed species, five were rated high, eight were rated medium, and seven were rated low for RDE prioritization (Table 2). Therefore, the objective of evaluating species with a sufficiently wide range of total resultant scores and therefore range of RDE prioritization was met.

Much of the information needed to address the 10 questions for each of the 20 weed species was available in the literature, so that expert opinion was rarely required where knowledge gaps existed. In this study, there are numerous references of government or industry online extension bulletins in addition to primary (original source) publications. This mix of references was a deliberate strategy as a lay person looking for information to address the 10 questions of AWAC for a species of interest would first peruse these readily available and plain language online extension resources.

*Lolium rigidum* scored 70 out of a possible 100 points. Therefore, the RDE priority was rated high. This species was viewed as a control treatment in AWAC, as a high score or rating was originally expected. Investment in RDE for this weed species has been consistently prioritized over the past 40 years, as evident from the numerous research studies conducted across Australia during this period. *Lolium rigidum* densities up to 300 plants m^-2^ can cause cereal crop yield losses of up to 75% [12,43]. It causes annual crop revenue (yield x price) losses of nearly AUD 100 million (revenue losses for surveyed species compiled in Llewellyn et al. [9]). Because of the prevalence and economic importance of the species across the grain-growing regions of Australia, it is often the “driver” weed in terms of dictating a grower’s herbicide program. Resistance in *L. rigidum* populations to multiple herbicide SOA is widespread across the Australian grain-growing region [34,37,44,45,46]. This allogamous weed is capable of long-distance (kilometre scale) pollen movement, which has aided herbicide resistance allele dispersal and hindered management of the weed across the landscape [47]. With high incidence of resistance to the main post-emergence herbicides, ACCase and ALS inhibitors, growers are increasingly relying upon soil-applied pre-emergence herbicides, such as trifluralin, propyzamide, prosulfocarb, and pyroxasulfone to control *L. rigidum* in cereal, oilseed, or pulse crops [48]. Livestock grazing or various non-selective and selective herbicides are employed to control the weed in fallow or pasture phases [49]. *Lolium rigidum* inflorescences (spikes) infested with the pathogen *Corynebacterium* or *Clavibacter* sp. residing in galls caused by nematodes (*Anguina* sp.) can cause livestock poisoning (“annual ryegrass toxicity”) due to the ingestion of corynetoxins [50,51]. *L. rigidum* is also susceptible to ergot fungus (*Claviceps purpurea*); the contaminated grain is downgraded and can result in toxicity to mammals when ingested [14].

*Raphanus raphanistrum* had the same total score as *L. rigidum*, i.e., 70 points (Table 2). Moreover, its individual scores matched those of *L. rigidum*. In the western and southern regions, *R. raphanistrum* has been the top broadleaf weed species targeted for RDE because of the same reasons described above for *L. rigidum*—high population abundance, crop yield or quality loss, and herbicide cost. In Australia, it is the most economically damaging broadleaf weed of the major grain crops (annual revenue loss of AUD 53 million). Effective herbicide control is critical, as even low plant densities can result in significant crop yield loss [52,53]. Continued high population abundance is partially due to high prevalence of herbicide resistance. Herbicide resistance in *R. raphanistrum* populations is widespread in Western Australia [35]. In a random survey in the Western Australian grain belt in 2015, 88% of populations were ALS inhibitor-resistant, 65% were phytoene desaturase (PDS) inhibitor (diflufenican)-resistant, 61% were auxinic (2,4-D)-resistant, and 14% were photosystem-II inhibitor (atrazine)-resistant [46]. Of concern is the widespread occurrence of multiple-resistant populations (combinations of ALS inhibitor, synthetic auxin, PDS inhibitor, photosystem-II inhibitor). In a 2003 random survey across the state, 58% of populations were resistant to two SOA, 18% to three SOA, and 7% to four SOA [54]. The most prevalent multiple resistance pattern was ALS inhibitor plus synthetic auxin. Pollen flow of this obligate outcrossing weed contributes to herbicide resistance allele dispersal and seeds can be spread by wind and water [13,55]. Despite widespread herbicide resistance, there still are effective herbicide options for *R. raphanistrum* control in cereals, oilseeds, and pulses [55,56]. However, two herbicide applications are typically required within a growing season to effectively control the weed [57], which increases herbicide costs. The species is considered a contaminant in cereal and canola seedlots because of similar seed size or pod breakage hindering effective crop seed cleaning as well as release of toxic compounds by pods and seeds [14].

The *Avena* spp. (*A. fatua* and *A. sterilis*) had a total score of 50 points and therefore rated medium for RDE prioritization (Table 2). Both *Avena* spp. rank high in terms of their competitive ability and can significantly reduce crop yields by up to 70% (reviewed in Bajwa et al.; Beckie et al. [58,59]). *Avena* spp. rank third in revenue loss in Australia (AUD 28 million). However, these *Avena* spp. are generally not considered a serious contaminant of the major grain crops. They are highly selfing, with limited natural seed dispersal capability. Herbicides of different SOA are available to control *Avena* spp., although commonly used herbicides are post-emergence ACCase or ALS inhibitors. Consequently, incidence of ACCase-inhibitor resistance is becoming more prevalent in *Avena* spp. populations across the country [46,60]. Nevertheless, alternative herbicides are available for their control in cereal, oilseed, or pulse crops. The herbicide cost to control *Avena* spp. is similar to other major grass weed species in Australia; in 1999, this cost to the wheat industry alone was estimated at AUD 60 million [14].

The *Hordeum* spp. (*H. leporinum* and *H. glaucum*) had a slightly higher total score (60) than that of *Avena* spp. (Table 2). The difference in scores was due to the trend of increasing relative abundance of *Hordeum* spp. over the past 20 years, particularly in the western and southern regions. Increasing abundance or range expansion reflects less effective control in crop or fallow, which may be impacted by the rising incidence of herbicide resistance. A number of populations of these species in the western and southern regions are resistant to ACCase inhibitors, ALS inhibitors, or photosystem-I disrupters (e.g., paraquat) [36].

*Hordeum* spp. can reduce wheat grain yield by 30% and economic returns by up to AUD 140 ha^−1^ [16]. There is a limited range of post-emergence herbicides available for their control in wheat and other cereals [14,61], and the herbicide cost per hectare is relatively high [32,62]. Similar to *Avena* spp., these *Hordeum* spp. are highly selfing with limited natural seed disperal. Moreover, they are not considered a contaminant of crop seedlots [61].

The total score assessed for *Bromus* spp. (*B. diandrus* and *B. rigidus*) was 80 and therefore rated high for RDE prioritization (Table 2). Although natural seed dispersal is limited, early seed shedding before crop harvest reduces the efficacy of harvest weed seed control practices and facilitates population persistence [18]. Another highly selfing plant [20], *Bromus* spp. are amongst the most competitive and economically damaging crop weeds in Australia [63]. In the same Western Australia study involving *Hordeum* spp, *Bromus* spp. reduced wheat grain yield by up to 43% and economic returns by up to AUD 205 ha^−1^ [16]. It causes annual crop revenue losses of AUD 23 million, the fourth highest among species. *Bromus* spp. are increasingly difficult to manage due to increased cropping intensity, lack of in-crop herbicides, herbicide resistance, and enhanced seed dormancy in some populations [18,64]. The widespread herbicide resistance in *Bromus* spp. results in an extra herbicide cost of AUD 3.2 million per year [9]. Populations have evolved resistance to ACCase inhibitors, ALS inhibitors, triazines, or glyphosate [36]. From 2010 to 2015, ALS inhibitor resistance in *Bromus* spp. increased from 12 to 24% of surveyed populations in Western Australia [46]. One-quarter of surveyed growers nationally cite *Bromus* spp. as their most costly weed to control [18]. Moreover, seeds cause contamination problems, for example, injury to livestock by entering the eyes, mouth, feet, and intestines [14].

*Arctotheca calendula* was evaluated as a medium RDE priority, with a total score of 55 (Table 2). It was not rated in the high category because of little indication of increasing abundance spatially or temporally, the availability of cost-effective herbicide options in crops, pastures, or fallow and limited extent of herbicide resistance. However, it remains a troublesome species for many farmers or land managers because of its high potential for crop interference, propagule spread, and toxicity to livestock. *Arctotheca calendula* can significantly reduce crop yields if not effectively controlled. In wheat, 7 to 90 plants m^-2^ can reduce grain yield by 28 to 44% [65]. The species is capable of efficient propagule disperal. Individuals may be transplanted during the planting operation, and seeds may be easily dispersed by wind, water, or animals [14]. The extent of herbicide resistance in *A. calendula* in Australia is limited to photosystem-I disrupters (e.g., paraquat, diquat) and auxinic herbicides [36]. Because *A. calendula* is an obligate outcrossing species, herbicide resistance may spread across the landscape via pollen movement [66]. The weed is associated with scouring in sheep and can cause nitrate and nitrite poisoning of livestock [14].

*Emex* spp. had a total score of 70, therefore rated as high for RDE prioritization (Table 2). Because it is considered as an emerging weed species, a better level of control that limits seed production in crop (especially pulses), pasture, or fallow land and better containment of natural and human-mediated seed spread are needed. The achenes (one-seeded fruits) can be easily dispersed by water (flotation) or livestock because of attachment by means of the sharp spines. *Emex* spp. causes economic losses of AUD 20 million annually over an estimated one million ha of cropland and one million ha of pastures in Western Australia alone; occurrence of eight to nine plants m^-2^ can reduce wheat yield by up to 50% [14,67]. There are a number of herbicide options to control the species in cereal crops, but not in pulse crops [68]. Some populations in Western Australia have evolved resistance to ALS-inhibiting herbicides [69]. The species is considered a contaminant because of the difficulty of separating achenes from the seeds of pulses; additionally, the plants are potentially toxic to livestock [70].

*Chloris virgata* tied *Bromus* spp. for the highest total score for RDE prioritization (80) (Table 2). An individual score for each question was greater than zero, except as a potential contaminant of major grain crops (question 8). Range expansion into the southern and western regions is concerning because of its many weediness traits and limited herbicide control options. *Chloris virgata* is a competitive, mostly wind-pollinated grass that requires an integrated and intensive management program for its control. Its abundance and persistence have been aided by no-tillage regimes and inherent tolerance or evolved resistance to glyphosate [36,71]. Pre-emergence and post-emergence herbicide options for *C. virgata* control in some winter or summer crops (e.g., sorghum, *Sorghum bicolor* (L.) Moench) and fallow are limited, therefore crop rotational planning is an important element of an integrated weed management program. The species can spread by seed dispersal and vegetatively by stolens [72]. It is important to control populations in ruderal areas such as roadsides and fence lines, since the weed can move into cropland by wind, water, or animals.

*Echinochloa* spp. were evaluated with a total score of 55, thereby rating medium for RDE prioritization (Table 2). These species rank fifth in annual crop revenue loss (AUD 15 million). *Echinochloa* spp. are problematic competitive weeds in sorghum, mung bean (*Vigna radiata* (L.) R. Wilczek), maize (*Zea mays* L.), rice (*Oryza sativa* L.), cotton (*Gossypium hirsutum* L.), and other summer field crops in the northern region. For example, these weed species can reduce sorghum yields by 25 to 40% [23]. Effective herbicides to control these species may not be available in all crops in the northern region. Herbicide resistance, particularly to glyphosate, is becoming widespread in *Echinochloa* spp. in Australia [36]. Therefore, glyphosate should not be used alone to control these species in fallow situations or as a preseeding burndown treatment. Because of the competitive nature of these two species and increasing incidence of populations with herbicide resistance, more research is needed on the efficacy of integrated cultural weed management practices to reduce their abundance and to reduce overall herbicide selection pressure [73].

Similar to *Echinochloa* spp., the total score for *Sonchus oleraceus* was 50 (medium rating) (Table 2). Monitoring of this weed across Australia is warranted given its continuing high (northern region) or increasing abundance (southern and western regions). Proliferation of *S. oleraceous* in low disturbance no-tillage regimes, increasing herbicide selection pressure and consequently herbicide resistance (especially glyphosate), and efficient seed dispersal will challenge future management. Although not generally a strong crop competitor [16], *S. oleraceus* is a serious weed in the northern region requiring herbicidal control in both winter and summer crops and fallow [74]. The small light seeds (achenes with pappus) are easily dispersed by wind [24]. Increasing incidence of resistance to ALS-inhibiting herbicides, auxinic herbicides, or glyphosate in the northern and southern regions [36,60] has impacted herbicidal control of the species.

A similar assessment (medium rating) was made for *Conyza* spp., based on a total score of 60 (Table 2). *Conyza* spp. (mainly *C. bonariensis*) are summer annual weed species commonly found in fallow land and can be difficult to consistently control with herbicides largely due to variable growth stage [14,75]. *Conyza bonariensis* can reduce sorghum yields by 60 to 100% [76]. Widespread resistance to glyphosate and some other herbicides further control efficacy and increase costs [36,77]. Similar to *S. oleraceus*, the pappus on the seed enables the species to be easily and rapidly dispersed long distances by wind. The goal of reducing the species’ abundance and economic impact is best achieved using a coordinated regional approach, similar to that recommended for other highly mobile weeds [39]. Such an approach is exemplified by a landscape (crop and non-crop area) monitoring and management project recently established in eastern Australia [78].

*Lactuca serriola* was assessed a total score of 40, therefore a medium rating (Table 2). Similar to *Conyza* spp. and *Sonchus oleraceus*, the light-weight seeds with pappus are easily dispersed by wind and through surface water run-off; for example, Lu et al. [79] reported that the seed can travel over distances up to 43 km. Consequently, the spread of herbicide resistance alleles over long distances is possible and has been quantified [79]. *Lactuca serriola* can grow up to 2 m high and is highly competitive with crops or pastures [28]. Moreover, flowering buds of this self-pollinated weed are often admixed with crop seed during harvest and difficult to screen out, resulting in seedlot contamination and reduced economic value [80]. In southern Australia, the species has evolved resistance to ALS inhibitors or glyphosate [36,79].

The same rating (score of 40) was assessed for *Sisymbrium orientale*, based in part on its potential adverse economic impact on winter crop yield and quality (Table 2). *Sisymbrium orientale* is another self-pollinated weed that is abundant across southern Australia and can cause substantial yield and quality losses in some winter crops [81]. Additionally, the small seeds can contaminate grain and therefore result in a potential grade reduction [14]. There is widespread incidence of ALS inhibitor resistance in populations in the southern region; moreover, resistance to three other herbicide SOA have been reported in populations in Australia [36,82].

Remaining evaluated weed species were rated low for RDE prioritization, each with a total score of less than 40 (Table 2). Of all the species rated low for RDE prioritization, *Rapistrum rugosum* scored the highest (35) because of its high relative abundance, crop competitiveness, and as a crop seed contaminant. *Rapistrum rugosum* is a competitive weed, causing crop yield reductions of up to 50%; it is a contaminant of grain seedlots, having a maximum threshold content [14]. A number of populations are reported as resistant to ALS-inhibiting herbicides [36].

Four species had a score of 30. This score for *Fallopia convolvulus* and *Chondrilla juncea* was mainly due to its potential impact on grain crop yield or quality. *Fallopia convolvulus*, a self-pollinated weed, can cause significant crop yield loss when present at high plant densities; the weed is also a contaminant of grain samples because of the difficulty in effectively removing its seeds [83]. Upper-limit thresholds of weed seed content have been established in milling grades of wheat [14]. There are reports of some populations in Queensland with resistance to ALS-inhibiting herbicides [36]. However, incidence of herbicide resistance in this species in Australia is limited. *Chondrilla juncea* can reduce crop yield by up to 80% [84,85]. Wind-mediated, pappus-assisted seed dispersal allows average movement of 24 km per year [84,86].

For *Polygonum* spp., this score was derived largely from its current and increasing relative abundance and because of its status as a contaminant. *Polygonum* spp. are moderately competitive species; in a study in Western Australia, *Polygonum* spp. reduced wheat grain yield by 12%, with an economic loss of AUD 57 ha^−1^ [16]. Furthermore, the species can be toxic to livestock [14,87]. The score of 30 for *Moorochloa eruciformis* was attributed to its increasing occurrence in northern grain-cropping areas, moderate degree of crop competitiveness, and limited incidence of herbicide resistance. Similar to the level of crop interference of *Polygonum* spp., *Moorochloa eruciformis* can reduce sorghum yields by 10 to 20%; it is most competitive when it forms dense patches that impede crop emergence and growth [14]. In fallow land in Queensland, populations of the weed were reported to be resistant to glyphosate in 2014 [36]. The total score for *Amsinckia* spp. was 20, which comprised individual scores for crop competitiveness and crop contamination. Similar to *R. rugosum*, *Amsinckia* spp. can cause significant cereal crop yield loss and is a contaminant of grain seedlots. Seeds of the species contain poisonous alkaloids that are toxic to some livestock [29]. Of all the weeds evaluated, *Vicia* spp. in southern Australia had the lowest total score (15). Although cultivated for fodder production and as a cover crop, *Vicia* spp. can cause significant grain crop revenue loss [9]. Depending upon the particular species, it may be a potential contaminant of crop seeds [88].

**Table 2 plants-09-01737-t002:** Individual and total scores from the Agricultural Weed Assessment Calculator (AWAC) for species in the western (W), southern (S), or northern (N) grain-cropping regions of Australia (rating categories of high (H): 70 to 100 points; medium (M): 40 to < 70 points; low (L): <40 points). The 10 questions are listed in Table 1. For question 1, ranking is on a national basis across the three regions.

Question	1	2	3	4	5	6	7	8	9	10	Total	Rating	Region	References
Species														
*Lolium rigidum*	10	0	0	0	10	10	10	10	10	10	70	H	WSN	[9,12,14,34,37,43,44,45,46,47,48,49,50,51]
*Bromus* spp.	10	10	10	0	10	10	0	10	10	10	80	H	WS	[9,14,16,18,19,20,36,46,63,64]
*Emex* spp.	5	10	10	0	5	10	10	10	10	0	70	H	WSN	[9,14,16,21,67,68,69,70]
*Chloris virgata*	5	10	10	10	5	10	10	0	10	10	80	H	NS	[9,22,36,71,72]
*Raphanus raphanistrum*	10	0	0	0	10	10	10	10	10	10	70	H	WSN	[9,13,14,35,46,52,53,54,55,56,57]
*Echinochloa* spp.	5	10	0	0	10	10	0	0	10	10	55	M	N	[9,14,23,36,73]
*Hordeum* spp.	10	10	0	0	10	10	0	0	10	10	60	M	WS	[9,14,16,17,32,36,61,62]
*Arctotheca calendula*	10	0	0	0	5	10	10	10	10	0	55	M	WSN	[9,14,19,36,65,66]
*Conyza* spp.	10	10	0	0	10	10	10	0	10	0	60	M	WSN	[8,9,14,27,36,39,75,76,77,78]
*Lactuca serriola*	0	0	0	0	10	10	10	10	0	0	40	M	S	[9,28,36,79,80]
*Sonchus oleraceus*	10	10	0	0	10	0	10	0	10	0	50	M	WSN	[9,14,16,24,25,26,36,60,74]
*Avena* spp.	10	0	0	0	10	10	0	0	10	10	50	M	WSN	[9,14,15,46,58,59,60]
*Sisymbrium orientale*	10	0	0	0	10	10	0	10	0	0	40	M	WSN	[9,14,36,81,82]
*Fallopia convolvulus*	5	0	0	0	5	10	0	10	0	0	30	L	N	[9,14,36,83]
*Amsinckia* spp.	0	0	0	0	0	10	0	10	0	0	20	L	WS	[9,29]
*Chondrilla juncea*	0	0	0	0	0	10	10	0	10	0	30	L	WS	[9,84,85,86]
*Moorochloa eruciformis*	0	10	0	0	5	5	0	0	10	0	30	L	N	[9,14,36]
*Rapistrum rugosum*	10	0	0	0	5	10	0	10	0	0	35	L	WSN	[9,14,36]
*Vicia* spp.	5	0	0	0	5	5	0	0	0	0	15	L	S	[9,88]
*Polygonum* spp.	5	10	0	0	0	5	0	10	0	0	30	L	WSN	[9,14,16,87]

One weed species examined in this study exemplifies how AWAC may evaluate overall risk differently than the Australian weed risk assessment system. The perennial weed, *Chondrilla juncea*, was rated low in AWAC, even though millions of dollars have been spent on biosecurity programs in Australia to contain this important invasive weed as rated by the Australian weed risk assessment system. It is precisely those proactive biosecurity programs that have kept the weed from increasing its range and abundance in cropland. AWAC does not focus on the environmental impacts of species, where the principal concern is the potential for displacement of native plant species. As noted in the Introduction, all of the 20 weeds examined in this study are alien species. However, a number of important agricultural weeds in Australia are native species, such as *Salsola australis* R.Br.

The utility of AWAC depends on robust and current weed abundance data as well as information on the economic impact of the species on crop production. In some regions of Australia, recent field survey data of the distribution and abundance of agricultural weeds is lacking. For some emerging weeds to be adequately assessed by AWAC, there may be insufficient information to address all of the 10 questions. Missing information for a species may be extrapolated from other agroregions of the world where it occurs. Emerging weed species will need to be continually evaluated as new information becomes available on its biology, distribution and abundance, and management.

Overall, scores were consistent with the current state of knowledge of the species’ impact on grain crop production in Australia. The AWAC still requires testing by weed practitioners (e.g., agronomists, consultants) from across Australia to verify its utility and robustness in rating these weed species and others not included in this initial evaluation. Its applicability to agricultural weeds in other countries is uncertain, although the questions constituting AWAC are relatively generic and not restricted to particular climatic or environmental conditions. Future refinements to AWAC are expected based on testing feedback. Following completion of the testing phase, it may become a useful decision-support tool for public- or private-sector organizations involved in supporting or conducting agricultural weed RDE activities.

## Figures and Tables

**Table 1 plants-09-01737-t001:** Agricultural Weed Assessment Calculator (AWAC): research, development, and extension prioritization of winter or summer weeds in the western, southern, or northern grain-cropping regions of Australia (rating categories of high: 70 to 100 points; medium: 40 to <70 points; low: <40 points).

AWAC Questionnaire
Does the weed species rank high (vs. other weeds) in relative abundance in the designated region? Ranking may be based on field survey incidence and abundance, grower surveys or feedback, herbarium data, etc. (ranks within the top 10 species: 10 points; ranks below the top 10, but within the top 20 species: 5 points; ranks below the top 20 species: 0 points) Has the weed species consistently increased in relative abundance over the last 20 years in the designated region? (large increase (by >30%): 10 points; moderate increase (by <30%): 5 points; no increase: 0 points) Are there registered herbicide options (pre- or post-emergence) to control the weed species in the major crops grown in the designated region? (no crops: 10 points; only some crops: 5 points; yes, all crops: 0 points) Are there herbicide or non-herbicide options to control the weed species during the non-crop phase (for example, fallow or pasture) in the designated region? (no: 10 points; yes: 0 points) What is the extent of herbicide resistance in this weed species in the designated region based on field surveys or grower sample testing? (high or widespread, i.e., >30% of populations or fields: 10 points; moderate or limited, i.e., 10–30% of populations or fields: 5 points; low or little (<10% of populations or fields: 0 points). Can the weed species potentially cause significant crop yield loss if left uncontrolled? (potential yield loss of: >30%: 10 points; 10–30%: 5 points; <10%: 0 points) Does the weed species have high potential for propagule (seed, pollen, or vegetative) dispersal? (yes: 10 points; no: 0 points) Is the weed species considered a contaminant of any major grain crops, for example, difficult to separate its seeds from crop seeds; potentially toxic to livestock, etc.? (yes: 10 points; no: 0 points) Is the weed species generally considered troublesome (economically damaging or difficult and costly to control) by growers in the designated region? (yes: 10 points: no: 0 points) Is the cost to control the weed species over the growing season in the designated region greater, similar, or lower compared with that of other weeds generally or on an absolute basis? (greater or similar: 10 points; lower: 0 points)

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
