# Peer review of "Agricultural Weed Assessment Calculator: An Australian Evaluation"

_plants, 2020, doi:10.3390/plants9121737_

Round 1

Reviewer 1 Report

Below, some general suggestions for improving the structure of the manuscript are provided.

  • The objective of the work, although it can be understood in several parts of the text, is not immediately clear. I suggest describing it more clearly at the end of the introduction.
  • Another information that should be shown more clearly and schematically are the bibliographic sources on the basis of which the score is assigned to each species. This information (in form of references) could be listed in the Table 2.
  • Latin names of the species should be written in Italics

Author Response

Reviewer 1 - replies indicated in italics

Below, some general suggestions for improving the structure of the manuscript are provided.

  • The objective of the work, although it can be understood in several parts of the text, is not immediately clear. I suggest describing it more clearly at the end of the introduction.

The objective is now described more clearly at the end of the introduction

  • Another information that should be shown more clearly and schematically are the bibliographic sources on the basis of which the score is assigned to each species. This information (in form of references) could be listed in the Table 2.

This is now included in the table

  • Latin names of the species should be written in Italics

We have checked that all latin names are now italicized

Reviewer 2 Report

Manuscript reviewed:

Plants-1032393; Agricultural weed assessment calculator: an Australian assessment

Subject:

This manuscript describes and evaluates a process for ranking introduced undesirable agricultural plant species in relation to a score derived from a series of questions associated with abundance, economic costs of control, and losses related to their presence in crops or products. The species scores provide a basis for prioritizing research budgets and extension information. The 20 species evaluated are described and ranked, and the resulting scores are assessed for appropriate values.

Overall, the manuscript very well organized and written, and I found only several questions of sentence structure that I identified (see below). Although the manuscript was on a topic with information that might be appropriate on a herbicide label for use and restrictions, I found the subject matter very informative and presented in an interesting and easily understandable style of writing. The authors have referenced, extracted and presented appropriate information from a large number of sources. Perhaps because of my personal interests in this topic, as a non-Australian I found the biological and ecological information presented with the subject species very informative and educational. This manuscript is one of the best written and organized that I have reviewed recently.    

The term "weed" in various forms appears >200 times. Intuitively, I suggest that each reader knows a definition of that term, but perhaps the author's definition would be useful. Line 34 appears to provide that definition as an introduced alien plant with adverse impacts. I guess my comment is related to native species; can some of them be considered as weeds? In a tree seedling nursery setting, native plants other than the cultivated species could be considered weeds. Having said this, I suppose readers will know what the authors intend "weed" to mean without further definition.

Detailed reading of the manuscript revealed only three minor items of text for the authors to consider for revision or clarification:

L319 Should "can reduced" be changed to "can reduce?"

L422 A word or words seems to be missing from "... and therefore a potential grade reduction." A revision might be "... and therefore result in a potential grade reduction."

L432 In the sentence beginning "Fallopia convolvulus ... "weed" is used three times. Although the sentence is suitable as presented, would it be improved if the last occurrence of weed at "... removing the weed seeds" could weed be replaced with “its" to read "... removing its seeds."

Author Response

Reviewer 2 -replies indicated in italics

This manuscript describes and evaluates a process for ranking introduced undesirable agricultural plant species in relation to a score derived from a series of questions associated with abundance, economic costs of control, and losses related to their presence in crops or products. The species scores provide a basis for prioritizing research budgets and extension information. The 20 species evaluated are described and ranked, and the resulting scores are assessed for appropriate values.

Overall, the manuscript is very well organized and written, and I found only several questions of sentence structure that I identified (see below). Although the manuscript was on a topic with information that might be appropriate on a herbicide label for use and restrictions, I found the subject matter very informative and presented in an interesting and easily understandable style of writing. The authors have referenced, extracted and presented appropriate information from a large number of sources. Perhaps because of my personal interests in this topic, as a non-Australian I found the biological and ecological information presented with the subject species very informative and educational. This manuscript is one of the best written and organized that I have reviewed recently.    

The term "weed" in various forms appears >200 times. Intuitively, I suggest that each reader knows a definition of that term, but perhaps the author's definition would be useful. Line 34 appears to provide that definition as an introduced alien plant with adverse impacts. I guess my comment is related to native species; can some of them be considered as weeds? In a tree seedling nursery setting, native plants other than the cultivated species could be considered weeds. Having said this, I suppose readers will know what the authors intend "weed" to mean without further definition.

You are correct – a ‘weed’ can certainly include and indeed does include native species, although most weed risk assessment systems target alien species. We have added a sentence (line 512 of revised version) to clarify this point.

“However, a number of important agricultural weeds in Australia are native species, such as Salsola australis R.Br.”

Detailed reading of the manuscript revealed only three minor items of text for the authors to consider for revision or clarification:

L319 Should "can reduced" be changed to "can reduce?"

Yes; Corrected

L422 A word or words seems to be missing from "... and therefore a potential grade reduction." A revision might be "... and therefore result in a potential grade reduction."

Revised as suggested

L432 In the sentence beginning "Fallopia convolvulus ... "weed" is used three times. Although the sentence is suitable as presented, would it be improved if the last occurrence of weed at "... removing the weed seeds" could weed be replaced with “its" to read "... removing its seeds."

Yes; revised as suggested
